# Mechanical Cavity Creation with Curettage and Vacuum Suction (Q-VAC) in Lytic Vertebral Body Lesions with Posterior Wall Dehiscence and Epidural Mass before Cement Augmentation

**DOI:** 10.3390/medicina55100633

**Published:** 2019-09-24

**Authors:** Eike I. Piechowiak, Maurizio Isalberti, Marco Pileggi, Daniela Distefano, Joshua A. Hirsch, Alessandro Cianfoni

**Affiliations:** 1University Institute of Diagnostic and Interventional Neuroradiology, Inselspital, Bern University Hospital, 3010 Bern, Switzerland; Alessandro.Cianfoni@eoc.ch; 2Department of Neuroradiology, Neurocenter of Southern Switzerland, 6900 Lugano, Switzerland; Maurizio.Isalberti@eoc.ch (M.I.); Marco.Pileggi@eoc.ch (M.P.); Daniela.Distefano@eoc.ch (D.D.); 3Department of Radiology, Massachusetts General Hospital, Harvard Medical School, Boston, MA 02114, USA; jahirsch@mgh.harvard.edu

**Keywords:** vertebral augmentation, cavity creation, lytic vertebral body lesions, vertebral body stent

## Abstract

*Background and Objectives:* We describe a novel technique for percutaneous tumor debulking and cavity creation in patients with extensive lytic lesions of the vertebral body including posterior wall dehiscence prior to vertebral augmentation (VA) procedures. The mechanical cavity is created with a combination of curettage and vacuum suction (Q-VAC). Balloon kyphoplasty and vertebral body stenting are used to treat neoplastic vertebral lesions and might reduce the rate of cement leakage, especially in presence of posterior wall dehiscence. However, these techniques could theoretically lead to increased intravertebral pressure during balloon inflation with possible mobilization of soft tissue tumor through the posterior wall, aggravation of spinal stenosis, and resultant complications. Creation of a void or cavity prior to balloon expansion and/or cement injection would potentially reduce these risks. *Materials and Methods*: A curette is coaxially inserted in the vertebral body via transpedicular access trocars. The intravertebral neoplastic soft tissue is fragmented by multiple rotational and translational movements. Subsequently, vacuum aspiration is applied via one of two 10 G cannulas that had been introduced directly into the fragmented lesion, while saline is passively flushed via the contralateral cannula, with lavage of the fragmented solid and fluid-necrotic tumor parts. *Results:* We applied the Q-VAC technique to 35 cases of thoracic and lumbar extreme osteolysis with epidural mass before vertebral body stenting (VBS) cement augmentation. We observed extravertebral cement leakage on postoperative CT in 34% of cases, but with no clinical consequences. No patients experienced periprocedural respiratory problems or new or worsening neurological deficit. *Conclusion:* The Q-VAC technique, combining mechanical curettage and vacuum suction, is a safe, inexpensive, and reliable method for percutaneous intravertebral tumor debulking and cavitation prior to VA. We propose the Q-VAC technique for cases with extensive neoplastic osteolysis, especially if cortical boundaries of the posterior wall are dehiscent and an epidural soft tissue mass is present.

## 1. Introduction

Percutaneous vertebral augmentation (VA) with vertebroplasty (PVP), balloon kyphoplasty (BKP), or vertebral body stenting (VBS) is often performed in patients with painful, fractured, or at-risk-of-fracture neoplastic spinal lytic lesions. The main goals are reinforcement of the vertebral body, stabilization or prevention of a fracture, and pain relief [1,2,3,4,5,6]. However, complication rate of VA, including cement pulmonary embolism and epidural cement leakage, is higher in patients with neoplastic when compared with osteoporotic fractures [7,8]. In spinal neoplastic lesions, the trabecular and spongious components of the vertebral body are infiltrated by tumoral tissue and the cortical boundaries might be eroded by neoplastic osteolysis; therefore, upon injection, the cement often distributes unevenly and unpredictably and has increased tendency to leak outside of the vertebral body [7]. The rate of cement leakage in metastatic lesions can reach up to approximately 70% [8]. While leakage rate might be reduced by the use of high-viscosity cement [9] and/or by balloon kyphoplasty [10], especially in presence of posterior wall lytic dehiscence and soft tissue epidural mass, both balloon expansion and cement injection might instead extrude tumoral tissue outside of the vertebral body, worsening a central canal stenosis [11] or facilitating extraosseous disease spread [12]. Moreover, raised intravertebral pressure during balloon inflation and cement injection has been shown to favor bone marrow and tumor cell migration in the systemic circulation [11,12,13,14], with demonstrated temporary raise of pulmonary arterial pressure [15,16], very rarely symptomatic, but with unknown clinical effects and impact on oncological outcome [1,17].

Creation of a void or cavity prior to balloon expansion and/or cement injection seems to lower intravertebral pressure, thereby facilitating a more secure filing of the lytic defect [18], and has the potential of reducing risk of cement leakage, soft tissue mass dislodgement, and pulmonary fat and neoplastic cells embolism. It ultimately allows a greater amount of cement deposition in the vertebral body.

Radiofrequency ablation (RFA) [14,19], cryoablation [20,21], coblation [20,21], curettage [22], and bone marrow washout [23], each with its own potential advantages and limits, have been proposed to decompress the vertebral body prior to cement injection.

The aim of this study is to describe a new percutaneous image-guided minimally invasive technique for mechanical nonthermal intravertebral tumor debulking and cavity creation in vertebral body lytic lesion. This technique, called “Q-VAC”, combines mechanical curettage and vacuum suction with lavage. We have applied this technique to cases with extensive osteolysis of the vertebral body, widely eroding cortical boundaries and posterior wall, often in the presence of an epidural mass, prior to VBS augmentation [24].

## 2. Materials and Methods

This is a technical note describing the procedural details and potential applications of this new technique, combining previously described and established procedures and devices [2,22,23,25]. We retrospectively evaluated all patients that underwent curettage and vacuum suction (Q-VAC) prior to cement augmentation at our institution between 01.03.2013 and 01.11.2018. Q-VAC technique was performed to aid in the cement augmentation of a spinal lytic metastatic lesion with extensive discontinuity of the cortical boundaries (“extreme osteolysis”). Since Q-VAC was performed with the intent to obtain satisfactory cement deposition in the vertebral body and to avoid undesired cement leakages and worsening of neurological status from tumoral soft tissue migration in the central canal, we considered the satisfactory stabilization of the lytic lesion as efficacy and any treatment-related clinical worsening due to cement leakage or tumor migration as complications. To assess the stability of the treated vertebral bodies, standing X-rays were obtained on the day following the procedure and 4 weeks after treatment.

The Institutional Review Board approved this investigation and the patients signed a required informed consent to undergo the procedure (Approval number: 2739 ID 14-136).

### Procedural Details

All interventions were performed in a mono- or biplanar angiography suite (Allura Xper, Philips, Best, The Netherlands). The patients were placed under general anesthesia while in the supine position and then turned into the prone position. Intravenous antibiotic prophylaxis was administered at the beginning of the procedure. After percutaneous fluoroscopically guided insertion of two 4.5 mm (7G) caliber trocars via transpedicular access (Access kit VBS, DePuySynthes-Johnson & Johnson, New Brunswick, NJ, USA), a Kyphon Latitude II Curette T-Tip 7 or 8 mm (Medtronic, Memphis, TN, USA) was coaxially inserted in the vertebral body via transpedicular access trocars. Subsequently, the curette tip was locked at 30, 60, or 90 degrees off-axis and the tissue present in the vertebral body was fragmented or “mashed-up” by multiple rotational (as a windshield wiper) and anteroposterior translational movements of the curette, while respecting the bony boundary of the vertebral body, under fluoroscopic control, until soft tissue consistency decreased due to tissue fragmentation. After retraction of the curette, a 10 G cannula was introduced into the now fragmented lesion via each access trocar. One cannula was connected to a 60-cc syringe filled with saline via a short luer-lock connection tubing and the second to a vacuum pump with a Penumbra Hi-Flow Aspiration Tubing (Penumbra, Alameda, CA, USA) producing aspiration force of 242 Mbar. The aspiration was then activated and the saline solution was passively flushed from the contralateral cannula through the fragmented lesion with lavage of the fragmented solid and fluid-necrotic tumor parts. Depending on the amount of tumor extraction or suspected residual tumor, repetition of the procedure was possible and performed at operator’s discretion. After cavity creation, insertion and expansion of the VBS, followed by cement augmentation, was performed as previously described [24]. Patients underwent postoperative plain films and CT and clinical follow-up, as reported in the clinical study [24]. Figure 1 shows an illustration of the technique.

Two illustrative cases present typical patients and treatments (Figure 2 and Figure 3).

## 3. Results

We applied the Q-VAC technique to 35 cases (19/16 M/F) (age 44–84, mean 67.9 y) of thoracic and lumbar (from T1 to L5) extreme osteolysis before VBS cement augmentation. Lytic lesions were related to solid tumor metastases in 27 cases and multiple myeloma in 8. In 21/35 cases, an extraosseous epidural mass was present on preprocedural imaging. We observed extravertebral cement leakage on postoperative CT in 34% of cases, but with no clinical consequences. No patients experienced periprocedural respiratory problems nor new or worsening neurological deficit. All treated vertebral bodies were stable at follow-up imaging, without secondary height loss.

## 4. Discussion

In patients with neoplastic lytic vertebral lesions, reducing pain, stabilizing fractures or lesions at risk of fracture, and ultimately improving quality of life are key elements of treatment. VA, with its technical variants, has an established role in achieving these goals [1,2,5]. However, not all procedures are applicable to extensive lytic lesions. VA of an extensive lesion with erosion of the posterior wall or epidural tumor spread bears the risk for spinal cord compression, either from cement leakage or from further central canal encroachment by the epidural mass, and risk of pulmonary cement embolism or tumor spread locally or hematogenously [8,11,13].

Creation of a cavity prior to cement injection or intravertebral device expansion, such as balloons and VBS, might help increase safety and avoid severe adverse events. There are alternatives which have been proposed to reduce the cement migration from vertebrae like BKP [2], RFA and cryoablation [26], and bone marrow lavage [23].

Specific limits of BKP concern the raise of intravertebral pressure provoked during balloons inflation that might displace tumoral tissue through a dehiscent posterior wall and cause further central canal stenosis, or mobilize bone marrow fat cells and neoplastic cellular aggregates into the systemic circulation. In addition, due to balloon deflation and its removal before cement injection, the intravertebral neoplastic tissue may re-expand elastically again, obliterating the previously created cavity.

RFA and cryoablation prior to cement injection result in reduction of tumor mass due to induction of necrosis [14,19,20,21]. These techniques can also cause thrombosis of the vertebral and paravertebral veins, reducing the PMMA embolization risk. Regarding the potential use to obtain an intravertebral cavity though, in both techniques, the induction of tumor cell necrosis does not correspond to an immediate void creation. Subsequent cement injection simply pushes residual tumor cells and necrosis aside. As additional drawbacks, RFA and cryoablation require a safety margin with vital and nervous structures, imply adjunctive time and cost increase [26].

Another described technique for cavity creation, the percutaneous controlled ablation (coblation), utilizes a plasma field to evaporate tumor cells at low temperatures, in theory allowing subsequent low-pressure cement injection with a reduced risk of cement leaks and epidural tumor displacement [3,27,28]. This technology, characterized by technical limitations in addressing large soft tissue lesions [24] and high costs, is no longer commercially available.

Although not truly a cavity creation technique, the bone marrow “washout” or lavage has been described firstly in a cadaveric spine model [29] and then in an animal model [30,31] to reduce cement injection forces, reducing cement extravasation, and fat embolic load to a degree below the threshold for eliciting a cardiovascular response. Jet lavage has also been reported in a clinical setting in a series of osteoporotic vertebral compression fractures [32] to potentially reduce the risk of cement leakage and prevent pulmonary embolism. Finally, bone marrow washout has been reported in a small series of patients treated with multilevel vertebroplasty for multiple myeloma spine lesions [23].

Nevertheless, we found simple aspiration or washout attempts is only able to partially remove the fluid, necrotic, or bloody parts of vertebral neoplastic lesions, as in multiple myeloma, but solid vertebral lesions commonly occur in metastatic breast and lung cancer cannot be removed with simple aspiration and lavage through transpedicular cannulas. For this reason, in the Q-VAC technique, before vacuum suction and lavage, we implemented a purely mechanical cavity creation using intravertebral soft tissue mass fragmentation through a curette. The use of a coaxial curette has been described in case of sclerotic changes after vertebral body fractures to maximize height restoration during balloon kyphoplasty [22], but it has not been employed to fragment neoplastic intravertebral soft tissue, nor combined to consequent aspiration.

Advantages of the Q-VAC are the creation of a true intravertebral cavity without increasing intraosseous pressure and without risk of thermal injuries to adjacent vital and nervous structures. The cavity creation adds no more than ten minutes to the procedure and does not require expensive devices. Nevertheless, the Q-VAC technique is only intended for creating a cavity in the vertebral body prior to VA, and by no means has the intent of local tumor control. Standardized oncological therapy should be considered as clinically indicated. Another limitation of this study is that it is single arm; Q-VAC has not been compared with any other debulking procedure.

Application: We propose the Q-VAC technique for cases with extensive neoplastic osteolysis of the vertebral body, especially if cortical boundaries of the posterior wall are dehiscent and an epidural soft tissue mass is present. In these cases, the Q-VAC allows a minimally invasive percutaneous debulking of the soft-tissue tumor component centrally located in the vertebral body, resulting in the creation of a cavity that in turn allows safer expansion of balloons if a BKP is to be performed or if VBS or stent-screw-assisted internal fixation (SAIF procedure) [24,25] is planned, and in general a potentially safer and more predictable deposition of larger amount of cement. In fact, in this severe lytic lesions, VBS and SAIF techniques offer a vertebral body reconstruction, with the stents acting as an internal scaffold representing a vertebral body prosthesis and helping contain the cement. Through these techniques, large volume implants are deployed and consequently large volume of cement can be deposited in the vertebral body, to the potential advantage of greater local stability. Preliminary cavity creation seems to be a desirable technical adjunct in such cases.

## 5. Conclusions

In our cohort, the Q-VAC technique, combining mechanical curettage and vacuum suction with lavage, is a safe, inexpensive, and reliable method for percutaneous intravertebral tumor debulking and cavitation prior to VA and its technical variants as VBS and SAIF, in extensive lytic vertebral neoplastic lesions.

## Figures and Tables

**Figure 1 medicina-55-00633-f001:**
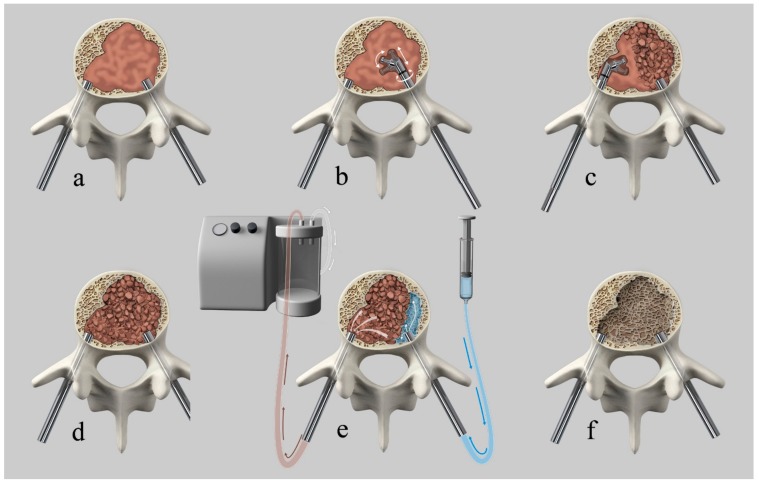
Illustration of the curettage and vacuum suction (Q-VAC) technique. (**a**) transpedicular introduction of the cannulas into the vertebral body. (**b**) coaxially inserted curette in the vertebral body via transpedicular access trocars with subsequent angulation of the curette and fragmentation of the solid lesion by multiple rotational and anteroposterior translational movements. (**c**) contralateral tumor fragmentation. (**d**) Illustration of the completely fragmented vertebral lesion. (**e**) connection of one cannula to a syringe filled with saline and the second to a vacuum pump. Activation of aspiration with subsequent passive flushing of saline through the fragmented lesion, with lavage of the fragmented solid and fluid-necrotic tumor parts. (**f**) created cavity after tumor debulking before subsequent vertebral augmentation.

**Figure 2 medicina-55-00633-f002:**
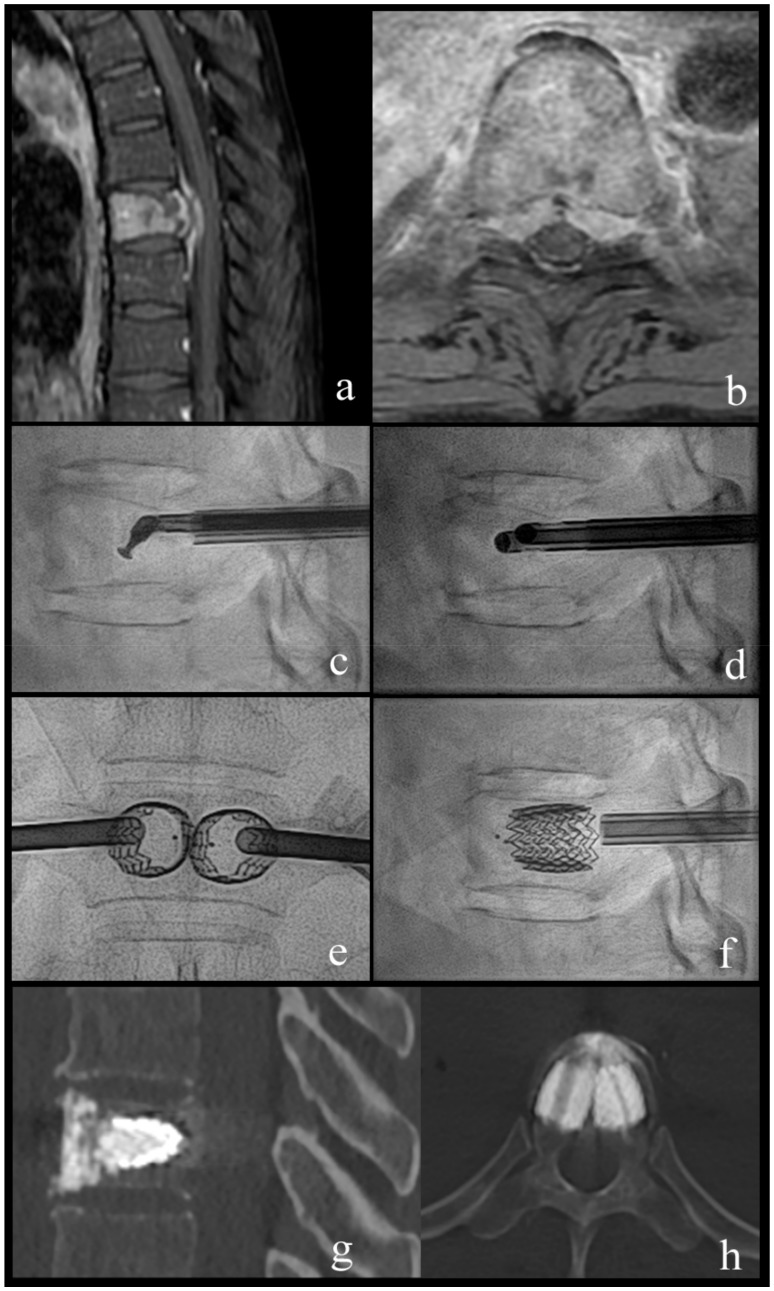
Case 1; a 63-year-old woman with breast cancer and newly diagnosed bone metastases. (**a**,**b**) sagittal and axial T1-weighted fat-suppressed enhanced MR images show vertebral lesion with involvement of the posterior wall and an epidural mass. (**c**) lateral fluoroscopy view with angulated coaxial curette in the vertebral body for lesion fragmentation and cavity creation. (**d**) lateral fluoroscopy view after introduction of two 10 G cannulas into the fragmented lesion for tumor flush and aspiration. (**e**,**f**) lateral and anteroposterior fluoroscopy views after vertebral body stenting (VBS) deployment with height restoration of the fractured vertebral body. (**g**,**h**) sagittal and axial CT after VBS and cement augmentation.

**Figure 3 medicina-55-00633-f003:**
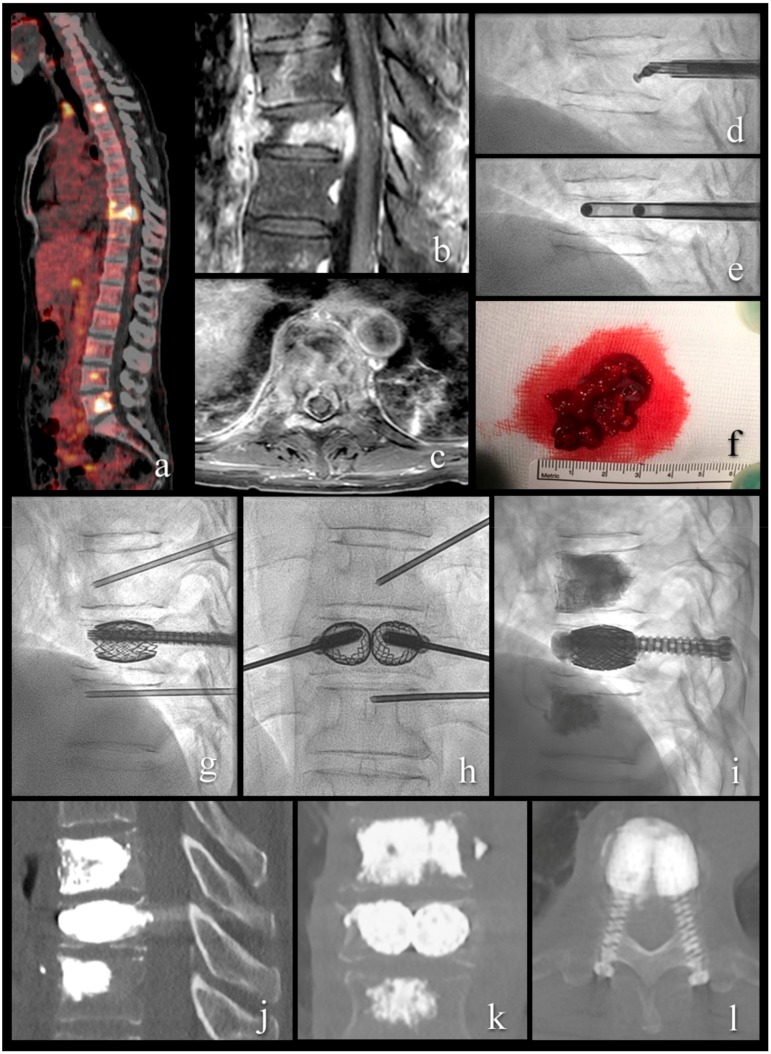
A 54-year-old patient with metastatic renal cell cancer and acute onset back pain. (**a**) FDG PET-CT with multiple spinal lesions with increased FDG uptake. (**b**,**c**) sagittal and axial T1-weighted fat-suppressed enhanced MR images show the vertebral lesion with involvement of the posterior wall, an epidural mass, and pathologic fracture. (**d**) lateral fluoroscopy view with angulated curette in the vertebral body for lesion fragmentation. (**e**) lateral fluoroscopy view after introduction of two 10 G cannulas into the fragmented lesion for tumor flush and aspiration. (**f**) aspirated tumor soft tissue, histologically compatible with renal cell cancer metastasis. (**g**–**i**) lateral and anteroposterior fluoroscopy views with stent-screw assisted internal fixation (SAIF) and cement augmentation. (**j**–**l**) sagittal, coronal, and axial CT after SAIF.

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
