# Peer review of "Mechanical Cavity Creation with Curettage and Vacuum Suction (Q-VAC) in Lytic Vertebral Body Lesions with Posterior Wall Dehiscence and Epidural Mass before Cement Augmentation"

_medicina, 2019, doi:10.3390/medicina55100633_

Round 1
Reviewer 1 Report
This is a nice technical note describing a novel technique to prepare vertebral body before cement augmentation. Description of procedure and illustrations are excellent and easy to follow. While this is not a scientific study, it would still benefit from a more detailed review of existing techniques and measures of success/failure/complications. In materials and methods, patient selection criteria, demographics, and pathology are all unclear. Results are very sparsely described, and it is unclear what measures should be considered post-procedure to evaluate the outcome / success. On a side note, it seems unclear if curetting is the best mechanical means to create vertebral cavity, or some other means (motorized, for example) would serve better?
Author Response
This is a nice technical note describing a novel technique to prepare vertebral body before cement augmentation. Description of procedure and illustrations are excellent and easy to follow.
While this is not a scientific study, it would still benefit from a more detailed review of existing techniques and measures of success/failure/complications.
Comment: We thank the reviewer for his kind comments. We discussed the advantages and disadvantages of the other available techniques in the discussion section, especially in relation to cavity creation. We have omitted a detailed in depth description of each technique, that we rather address to the relative References.
In materials and methods, patient selection criteria, demographics, and pathology are all unclear.
Comment: We agree with this comment and we added these information in the materials and methods section.
Results are very sparsely described, and it is unclear what measures should be considered post-procedure to evaluate the outcome / success.
Comment: We acknowledge this insightful comment and added these information in the materials and methods and results section. Since Q-Vac was performed with the intent to obtain satisfactory cement deposition in the vertebral body and to avoid undesired cement leakages and worsening of neurological status from tumoral soft tissue migration in the central canal, we considered as efficacy the satisfactory stabilization of the lytic lesion, and as complications any treatment-related clinical worsening due to cement leakage or tumor migration.
On a side note, it seems unclear if curetting is the best mechanical means to create vertebral cavity, or some other means (motorized, for example) would serve better?
Comment: This is an interesting question but we do not have the answer. It would need to be evaluated further, maybe in a comparative study. In this manuscript we explain our rationale behind the Q-Vac.
Reviewer 2 Report
The authors present an interesting study on treatment of lytic vertebral body lesions by using a new tool called Q-VaAC. It consists in the mechanical removal of neoplastic tissue and cavity creation on the vertebral body mechanical cavity followed by cement augmentation. The topic is interesting. The technique is quite novel. The study is simple and well written. It mainly consists in the description of the technique in 35 patients. Results are briefly reported and mainly consists in the absence of periprocedural respiratory problems nor new or worsening neurological deficits. Discussion is detailed and conclusions are supported by results. Figures are illustrative.
I have only a minor concern: why did the authors state the procedure is inexpensive? I do not think that statement is correct. Please, correct it or explain why the authors consider the procedure inexpensive.
Author Response
The authors present an interesting study on treatment of lytic vertebral body lesions by using a new tool called Q-VaAC. It consists in the mechanical removal of neoplastic tissue and cavity creation on the vertebral body mechanical cavity followed by cement augmentation. The topic is interesting. The technique is quite novel. The study is simple and well written. It mainly consists in the description of the technique in 35 patients. Results are briefly reported and mainly consists in the absence of periprocedural respiratory problems nor new or worsening neurological deficits. Discussion is detailed and conclusions are supported by results. Figures are illustrative.
I have only a minor concern: why did the authors state the procedure is inexpensive? I do not think that statement is correct. Please, correct it or explain why the authors consider the procedure inexpensive.
Comment: We thank the reviewer for his kind comments. Regarding the cost, we refer to the comparison of Q-vac with the other mentioned techniques. Compared to thermal ablation through RF or cryo, or to Coblation, the use of a simple manual curette and a vacuum aspiration as used in Q-Vac seems to be obviously less expensive in terms of ,aterials, and also it does not entail a significantly longer procedure.